# A Closer Look at Self-supervised Lightweight Vision Transformers

## Abstract

Self-supervised learning on large-scale Vision Transformers (ViTs) as pre-training methods has achieved promising downstream performance. Yet, how much these pre-training paradigms promote lightweight ViTs' performance is considerably less studied. In this work, we mainly develop and benchmark self-supervised pre-training methods, *e.g.*, contrastive-learning-based MoCo-v3, masked-image-modeling-based MAE on image classification tasks, and some downstream dense prediction tasks. We surprisingly find that if proper pre-training is adopted, even vanilla lightweight ViTs show comparable performance on ImageNet to previous SOTA networks with delicate architecture design. We also point out some defects of such pre-training, *e.g.*, failing to benefit from large-scale pre-training data and showing inferior performance on data-insufficient downstream tasks. Furthermore, we analyze and clearly show the effect of such pre-training by analyzing the properties of the layer representation and attention maps for related models. Finally, based on the above analyses, a distillation strategy during pre-training is developed, which leads to further downstream performance improvement for MAE-based pre-training.

## 1 Introduction

Self-supervised learning (SSL) has shown great progress in representation learning without heavy reliance on expensive labeled data. SSL focuses on various pretext tasks for pre-training. Among them, several works (He et al., 2020; Chen et al., 2020; Grill et al., 2020; Caron et al., 2020; Chen et al., 2021a; Caron et al., 2021) based on contrastive learning (CL) have achieved comparable or even better accuracy than supervised pre-training when transferring the learned representations to downstream tasks. Recently, another trend focuses on masked image modeling (MIM) (Bao et al., 2021; He et al., 2021; Zhou et al., 2022), which perfectly fits Vision Transformers (ViTs) (Dosovitskiy et al., 2020) for vision tasks, and achieves improved generalization performance. Most of these works, however, involve large networks with little attention paid to smaller ones. Some works (Fang et al., 2020; Abbasi Koohpayegani et al., 2020; Choi et al., 2021) focus on contrastive self-supervised learning on small convolutional networks (ConvNets) and improve the performance by distillation. However, the pre-training of lightweight ViTs is considerably less studied.

Efficient neural networks are essential for modern on-device computer vision. Recent study on achieving top-performing lightweight models mainly focuses on designing network architectures (Sandler et al., 2018; Howard et al., 2019; Graham et al., 2021; Ali et al., 2021; Heo et al., 2021; Touvron et al., 2021b; Mehta & Rastegari, 2022; Chen et al., 2021b; Pan et al., 2022), while with little attention on how to optimize the training strategies for these models. We believe the latter is also of vital importance, and the utilization of pre-training is one of the most hopeful approaches along this way, since it has achieved great progress on large models. To this end, we develop and benchmark recently popular self-supervised pre-training methods, *e.g.*, CL-based MoCo-v3 (Chen et al., 2021a) and MIM-based MAE (He et al., 2021), along with fully-supervised pre-training for lightweight ViTs as the baseline on both ImageNet and some other classification tasks as well as some dense prediction tasks, *e.g.*, object detection and segmentation. We surprisingly find that *if proper pre-training is adopted, even vanilla lightweight ViTs show comparable performance to previous SOTA networks with delicate design on ImageNet*, which achieves 78.5% top-1 accuracy on ImageNet with vanilla ViT-Tiny (5.7M). We also observe some intriguing defects of such pre-

training, *e.g.*, *failing to benefit from large-scale pre-training data* and *showing inferior performance on data-insufficient downstream tasks*.

These findings motivate us to dive deep into the working mechanism of these pre-training methods for lightweight ViTs. More specifically, we introduce a variety of model analysis methods to study the pattern of layer behaviors during pre-training and fine-tuning, and investigate what really matters for downstream performance. First, we find that *lower layers of the pre-trained models matter more than higher ones if sufficient downstream data is provided, while higher layers matter in data-insufficient downstream tasks*. Second, we observe that *the pre-training alters the attention behaviors of the final recognition model little, without introducing locality inductive bias*, which is, however, the commonly adopted rule for recent network architecture design (Mehta & Rastegari, 2022; Heo et al., 2021; Touvron et al., 2021b; Liu et al., 2021). Based on the above analyses, we also develop a distillation strategy for MAE-based pre-training, which improves the pre-training of lightweight ViTs. Better downstream performance is achieved especially on data-insufficient classification tasks and detection tasks.

## 2 PRELIMINARIES AND EXPERIMENTAL SETUP

**ViTs.**  We use ViT-Tiny (Touvron et al., 2021a) as the base model in our study to examine its downstream performance with pre-training, which contains 5.7M parameters. We adopt the vanilla architecture, consisting of 12 layers with the embedding dimension of 192, except that the number of heads is increased to 12 as we find it can improve the model's expressive power. We use this improved version by default. ViT-Tiny is chosen for study because it is an ideal experimental object, on which almost all existing pre-training methods can directly apply, and has a rather naive structure, which can eliminate the influence of the model architecture on our analysis to a great extent.

**Evaluation Metrics.**  *Linear probing* has been a popular protocol to evaluate the quality of the pre-trained weights (He et al., 2020; Chen et al., 2020; Grill et al., 2020; Caron et al., 2020), in which only the prediction head is tuned based on the downstream training set while the pre-trained representations are kept frozen. However, prior works point out that linear evaluation does not always correlate with utility (He et al., 2021; Newell & Deng, 2020).

*Fine-tuning* is another evaluation protocol, in which all the layers are tuned by first initializing them with the pre-trained models. We adopt this by default. Besides, layer-wise *lr* decay (Bao et al., 2021) is also taken into consideration. By default, we do the evaluation on ImageNet (Deng et al., 2009) by fine-tuning on the train split and evaluating on the validation split. Several other downstream classification datasets (Nilsback & Zisserman, 2008; Parkhi et al., 2012; Maji et al., 2013; Krause et al., 2013; Krizhevsky et al., 2009; Van Horn et al., 2018) and object detection and segmentation tasks on COCO (Lin et al., 2014) are also exploited for comparison in our study.

**Compared Methods.**  *Baseline*: We largely follow the recipe in DeiT (Touvron et al., 2021a) except for some hyper-parameters of augmentations (see Appendix A.1 for our improved recipe) and fully-supervised train a ViT-Tiny from scratch for 300 epochs on the training set of ImageNet-1k. It achieves 74.5% top-1 accuracy on the validation set of ImageNet-1k, surpassing that in the original architecture (72.2%) through modifying the number of heads to 12 from 3, and further reaches 75.8% by adopting the improved training recipe, which finally serves as our strong baseline to examine the pre-training. We denote this supervised trained model by DeiT-Tiny.

*MAE*: MAE (He et al., 2021) is selected as a representative for MIM-based pre-training methods, which has a simple framework with low training cost. We largely follow the design of MAE except that the encoder is altered to ViT-Tiny. Several basic factors and components are adjusted to fit the smaller encoder (see Appendix A.2). By default, we do pre-training on the train split of ImageNet-1k (Deng et al., 2009) (dubbed IN1K) for 400 epochs, and denote the pre-trained model as MAE-Tiny.

*MoCov3*: We also implement a contrastive SSL pre-training counterpart to achieve a more thorough study. MoCo-v3 (Chen et al., 2021a) is selected for its simplicity. We use MoCov3-Tiny to denote this pre-trained model with 400 epochs. Details are provided in Appendix A.3.

Some other methods, *e.g.*, MIM-based SimMIM Xie et al. (2022) and CL-based DINO Caron et al. (2021) are also involved, but are moved to Appendix B.5 due to the space limitation.

Table 1: **Comparisons on pre-training methods**. We report top-1 accuracy on the validation set of ImageNet-1k (Deng et al., 2009). IN1K and IN21K indicate the training set of ImageNet-1k and ImageNet-21k (Deng et al., 2009). The pre-training time is measured on 8×V100 GPU machine. ViT-Tiny is adopted for all entries. 'ori.' represents the training recipe in Touvron et al. (2021a) and 'impr.' represents our improved recipe (see Appendix A.1).

| | Pre-training | | | Fine-tuning | |
| Methods | Data | Epochs | Time (hour) | recipe | Top-1 Acc. (%) |
|---|---|---|---|---|---|
| from scratch | - | - | - | ori. | 74.5 |
| from scratch | - | - | - | impr. | 75.8 |
| Supervised (Steiner et al., 2021) | IN21K w/ labels | 30 | 20 | impr. | 76.9 |
| Supervised (Steiner et al., 2021) | IN21K w/ labels | 300 | 200 | impr. | 77.8 |
| MoCo-v3 (Chen et al., 2021a) | IN1K w/o labels | 400 | 52 | impr. | 73.7 |
| MAE (He et al., 2021) | IN1K w/o labels | 400 | 23 | impr. | **78.0** |

## 3 HOW WELL DOES PRE-TRAINING WORK ON LIGHTWEIGHT VITS?

**MAE outperforms other pre-training methods on ImageNet.** We develop and benchmark fully-supervised and self-supervised pre-training methods on ImageNet, as reported in Tab. 1. For all of the pre-trained models, we fine-tune them for 300 epochs on IN1k for fair comparisons. It can be seen that most of these supervised and self-supervised pre-training methods improve the downstream performance, whilst MAE outperforms others and consumes moderate training cost. Meanwhile the pre-training of MoCo-v3 leads to performance degradation. We denote the fine-tuned model based on the pre-training of MAE-Tiny as MAE-Tiny-FT.

**Enhanced vanilla ViTs with pre-training are comparable to previous SOTA networks.** We further compare the enhanced ViT-Tiny (5.7M) with MAE pre-training to the DeiT-Tiny (Touvron et al., 2021a) baseline and other previous lightweight ConvNets and ViT derivatives in Tab. 2. We report top-1 accuracy along with the model parameter count and the throughput, which is borrowed from PyTorch Image Models (timm) (Wightman, 2019). In specific, we extend the training epochs during fine-tuning to 1000 epochs following Touvron et al. (2021a). The resulting models are on par with or even outperform most previous ConvNets and ViT derivatives with comparable parameters or throughput. This demonstrates the usefulness of the advanced lightweight ViT pre-training strategy which is orthogonal to the network architecture design strategy in the ViT derivatives. Besides, we also compare with the methodology of pre-training lightweight ConvNets or ViT derivatives for a more fair comparison (Fang et al., 2020; Abbasi Koohpayegani et al., 2020; Gao et al., 2021; Choi et al., 2021). We find that most of them are evaluated under the *linear probing* protocol. We thus implemented the above methodology by ourselves, *i.e.*, adopting the pre-training with SEED (Fang et al., 2020) on EfficientNet-B0 (Tan & Le, 2019) under the *fine-tuning* protocol. The result, however, shows no improvement (from 77.7% to 77.2%).

**The pre-training benefits little from large-scale data.** Furthermore, we observe that MAE is robust to the pre-training dataset scale and class distribution in contrast to MoCo-v3 as shown in Tab. 3. We consider two subsets of IN1K containing 1% and 10% of the total examples (1% IN1K and 10% IN1K) balanced in terms of classes (Assran et al., 2021), one subset with long-tailed class distribution (Liu et al., 2019) (IN1K-LT), and IN21K. This observation

Table 3: **Effect of pre-training data**. Top-1 accuracy is reported.

| Datasets | MoCo-v3 | MAE |
|---|---|---|
| IN1K | 73.7 | 78.0 |
| 1% IN1K | 73.1 (-0.6) | 77.9 (-0.1) |
| 10% IN1K | 73.5 (-0.2) | 78.0 (+0.0) |
| IN1K-LT | 73.0 (-0.7) | 77.9 (-0.1) |
| IN21K | 73.8 (+0.1) | 78.0 (+0.0) |

is consistent with El-Nouby et al. (2021) on larger ViTs. It also reveals the limitation of these pre-training methods that they fail to benefit from large-scale pre-training data.

**Downstream data scale matters.** As shown in Tab. 4, we transfer the learned representations of different pre-trained models to several other downstream tasks (Nilsback & Zisserman, 2008; Parkhi et al., 2012; Maji et al., 2013; Krause et al., 2013; Krizhevsky et al., 2009; Van Horn et al., 2018) to investigate their effects. In addition to using the self-supervised pre-trained models, *i.e.*, MAE-Tiny and MoCov3-Tiny, both of which are pre-trained for 400 epochs, a fully-supervised counterpart based on IN1K with 300-epoch pre-training (*i.e.*, DeiT-Tiny) is also involved. An interesting observation is that the self-supervised pre-training approaches achieve downstream performance far

Table 2: **Comparisons with previous SOTA networks on ImageNet-1k.** We report top-1 accuracy on ImageNet-1k validation set (Deng et al., 2009), ImageNet Real (Beyer et al., 2020) and ImageNet V2 matched frequency (Recht et al., 2019), along with throughput and parameter count. The throughput is borrowed from timm (Wightman, 2019), which is measured on a single RTX 3090 GPU with a batch size fixed to 1024 and mixed precision. IN1K and IN21K indicate the training set of ImageNet-1k and ImageNet-21k. †indicates that distillation is adopted during the supervised training (or fine-tuning). ⋆ indicates the original architecture of ViT-Tiny, and others use the improved architecture (number of heads is changed to 12), *e.g.*, MAE-Tiny-FT in the table.

| Methods | pre-train data | #param. | throughput (image/s) | Val Top-1 | Real Top-1 | V2 Top-1 |
|---|---|---|---|---|---|---|
| *ConvNets* | | | | | | |
| ResNet-18 (He et al., 2016) | - | 12M | 8951 | 69.7 | 77.3 | 57.2 |
| ResNet-50 (He et al., 2016; Wightman et al., 2021) | - | 25M | 2696 | 80.4 | 85.7 | 68.7 |
| EfficientNet-B0 (Tan & Le, 2019) | - | 5M | 5369 | 77.7 | 84.0 | 66.3 |
| EfficientNet-B0 (Fang et al., 2020) | IN1K w/o labels | 5M | 5369 | 77.2 | 83.5 | 65.9 |
| EfficientNet-B1 (Tan & Le, 2019) | - | 8M | 2953 | 78.8 | 84.6 | 67.5 |
| MobileNet-v2 (Sandler et al., 2018) | - | 4M | 7909 | 72.0 | 80.2 | 60.2 |
| MobileNet-v3 (Howard et al., 2019) | - | 5M | 9113 | 75.2 | 82.2 | 63.4 |
| MobileNet-v3†(Beyer et al., 2021) | - | 5M | 9113 | 77.0 | - | - |
| *Vision Transformers Derivative* | | | | | | |
| LeViT-128 (Graham et al., 2021) | - | 9M | 13276 | 78.6 | 84.8 | 66.6 |
| LeViT-192 (Graham et al., 2021) | - | 11M | 11389 | 80.0 | 85.6 | 67.9 |
| XCiT-T12/16†(Ali et al., 2021) | - | 7M | 3157 | 78.6 | 84.1 | 67.0 |
| PiT-Ti†/ 1000 epochs (Heo et al., 2021) | - | 5M | 4547 | 76.4 | 82.0 | 63.1 |
| CaiT-XXS-24†(Touvron et al., 2021b) | - | 12M | 1351 | 78.4 | 85.2 | 67.4 |
| MobileViT-S (Mehta & Rastegari, 2022) | - | 6M | 1900 | 78.3 | 84.3 | 66.9 |
| Swin-1G (Liu et al., 2021; Chen et al., 2021b) | - | 7M | - | 77.3 | - | - |
| LVT (Yang et al., 2021) | - | 6M | - | 74.8 | - | - |
| EdgeViT-XS (Pan et al., 2022) | - | 7M | - | 77.5 | - | - |
| Mobile-Former-294M (Chen et al., 2021b) | - | 11M | - | 77.9 | - | - |
| *Vanilla Vision Transformers* | | | | | | |
| DeiT-Tiny⋆ (Touvron et al., 2021a) | - | 6M | 4844 | 72.2 | 80.1 | 60.4 |
| DeiT-Tiny⋆†/ 1000 epochs (Touvron et al., 2021a) | - | 6M | 4764 | 76.6 | 83.9 | 65.4 |
| MAE-Tiny-FT | IN1K w/o labels | 6M | 4020 | 78.0 | 84.3 | 66.2 |
| MAE-Tiny-FT / 1000 epochs | IN1K w/o labels | 6M | 3956 | 78.5 | 85.3 | 67.1 |

Table 4: **Transfer evaluation on classification tasks and dense-prediction tasks**. Self-supervised pre-training approaches generally show inferior performance to the fully-supervised counterpart. Top-1 accuracy is reported for classification tasks and AP is reported for object detection (det.) and instance segmentation (seg.) tasks. The description of each dataset is represented as (train-size/test-size/#classes).

| Datasets / Init. | Flowers (2k/6k/102) | Pets (4k/4k/37) | Aircraft (7k/3k/100) | Cars (8k/8k/196) | Cifar100 (50k/10k/100) | iNat18 (438k/24k/8142) | COCO(det.) (118k/50k/80) | COCO(seg.) |
|---|---|---|---|---|---|---|---|---|
| *supervised* | | | | | | | | |
| DeiT-Tiny | **96.4** | **93.1** | 73.5 | **85.6** | **85.8** | 63.6 | **40.7** | **36.5** |
| *self-supervised* | | | | | | | | |
| MoCov3-Tiny | 94.8 | 87.8 | **73.7** | 83.9 | 83.9 | 54.5 | 40.0 | 36.0 |
| MAE-Tiny | 85.8 | 76.5 | 64.6 | 78.8 | 78.9 | 60.6 | 38.9 | 35.1 |

behind the fully-supervised counterpart, while the performance gap is narrowed more or less as the data scale of the downstream task increases. Moreover, MAE even shows inferior results to MoCo-v3. We conjecture that it is due to their different layer behaviors during pre-training and fine-tuning, *e.g.*, undesired representations of the higher layers in MAE-Tiny, which will be discussed in detail in the following section. We refer the reader to Appendix A.4 for more details about those tasks.

For a more thorough study, we further evaluate on downstream object detection and segmentation tasks on COCO (Lin et al., 2014) based on Li et al. (2021) (see Appendix A.5 for details), with different pre-trained models as initialization of the backbone, as shown in Tab. 4. The self-supervised pre-training also lags behind the fully-supervised counterpart and MAE-Tiny still shows worse results than MoCov3-Tiny.

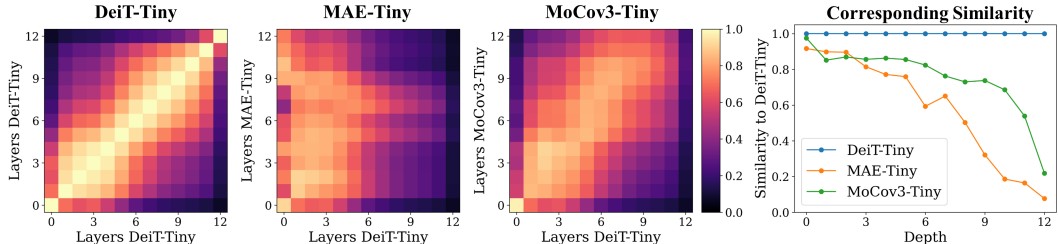

Figure 1: **Layer representation similarity** within and across models as heatmaps (the left three columns), with x and y axes indexing the layers (the 0 index indicates the patch embedding layer), and higher values indicate higher similarity. We also plot the corresponding layer similarity in the last column based on the diagonal elements of the left heatmaps.

# 4 REVEALING THE SECRETS OF THE PRE-TRAINING

In this section, we introduce some model analysis methods to study the pattern of layer behaviors during pre-training and fine-tuning, and investigate what matters for downstream performances.

## 4.1 LAYER REPRESENTATION ANALYSES

We first adopt Centered Kernel Alignment (CKA) method[1] (Cortes et al., 2012; Nguyen et al., 2020) to analyze the layer representation similarity across and within networks. Specifically, CKA computes the normalized similarity in terms of the Hilbert-Schmidt Independence Criterion (HSIC (Song et al., 2012)) between two feature maps or representations, which is invariant to the orthogonal transformation of representations and isotropic scaling (detailed in Appendix A.6). Fourier analysis of feature maps is also involved in analyzing the behaviors of the models.

**Lower layers matter more than higher ones if sufficient downstream data is provided.** We visualize the layer representation (Rep.) similarity between several pre-trained models and DeiT-Tiny as heatmaps in Fig. 1. The similarity within DeiT-Tiny is also presented for reference (the left column). We also plot the corresponding layer similarity in the last column based on the diagonal elements of the left heatmaps. We choose DeiT-Tiny as the reference because we consider the higher similarity between the pre-trained models and DeiT-Tiny (classification model fully-supervised trained from scratch) indicates more relevance to recognition for the self-supervised layers. Although the similarity does not directly indicate whether the downstream performance is good or not, it indeed reflects the pattern of layer representation to a certain extent. In Appendix B.1, stronger supervised trained ViTs are introduced as reference models, and we find that these supervised ViTs generally have similar layer representation structures.

First, We observe a relatively high similarity between MAE-Tiny and DeiT-Tiny for lower layers, while low similarity for higher layers. Similar phenomenon is observed w.r.t. other reference models as shown in Appendix B.1. It indicates fewer semantics are extracted for MAE-Tiny at a more abstract level in higher layers. Another empirical evidence is the low linear probing performance of MAE-Tiny (23.4% top-1 accuracy). In contrast, MoCov3-Tiny aligns DeiT-Tiny well across almost all layers. However, the fine-tuning evaluation in Tab. 1 shows that adopting the MAE-Tiny as initialization significantly improves the performance while MoCov3-Tiny degrades performance. Thus, we hypothesize that *lower layers matter much more than higher ones for the pre-trained models*. In order to verify the hypothesis, we design another experiment by only reserving several leading blocks of pre-trained models and randomly initializing the others, and then fine-tuning them on ImageNet (for the sake of simplicity, we

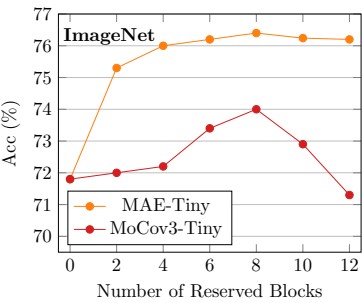

Figure 2: Lower layers of pre-trained models contribute to most gains on downstream ImageNet dataset.

domly initializing the others, and then fine-tuning them on ImageNet (for the sake of simplicity, we

---

[1]https://github.com/AntixK/PyTorch-Model-Compare

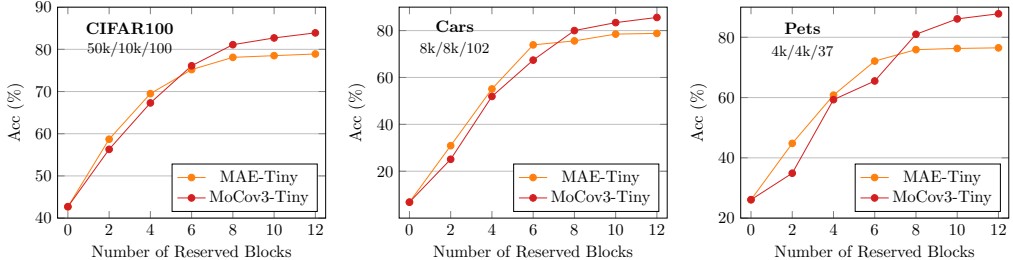

Figure 4: The contributions from higher layers on performance gain increase as the downstream dataset scale shrinks, which indicates higher layers matter in data-insufficient downstream tasks.

only fine-tune these models on IN1K for 100 epochs). Fig. 2 shows that reserving only a certain number of leading blocks achieves a significant performance gain over randomly initializing all the blocks (*i.e.*, totally training from scratch) for both MAE-Tiny and MoCov3-Tiny. Whereas, further reserving higher layers leads to marginal gain for MAE-Tiny or even degradation for MoCov3-Tiny, which demonstrates our hypothesis.

Then we examine why MoCov3-Tiny performs worse than MAE-Tiny. The Fourier analysis of feature maps is carried out as a supplement beyond CKA-based similarity, since we find that the similarity is largely dominated by low-frequency components. In Fig. 3, we plot the $\Delta$log amplitude across layers, which is the difference between the log amplitude at normalized low frequency ($0.0\pi$) and high frequency ($1.0\pi$) in each layer. It is also used in Park & Kim (2021) to analyze the differences between ViTs and ConvNets. We find that a large amount of high-frequency signals are reduced in the first layer of MoCov3-Tiny (*i.e.*, patch embedding layer), which shows a great difference from other models. This behavior possibly strengthens the robustness against various image augmentations, which is beneficial to the instance discrimination task, but results in an over-spatially-smoothed feature map at the very beginning of the network forward processing, leading to an inferior downstream performance on ImageNet.

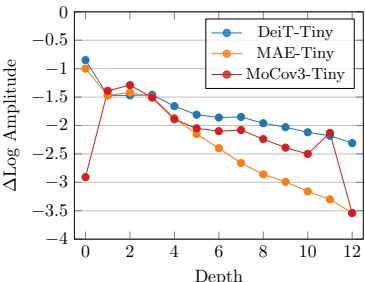

Figure 3: MoCov3-Tiny behaves differently at the first layer, reducing a lot of high-frequency signals.

**Higher layers matter in data-insufficient downstream tasks.** Previous works (Touvron et al., 2021a; Raghu et al., 2021) demonstrate the importance of a relatively large dataset scale for high-performance ViTs with large model sizes. We also observe a similar phenomenon on lightweight ViTs even with the self-supervised pre-training adopted as discussed in Sec. 3. It motivates us to study the key factor in downstream performance on data-insufficient tasks.

We conduct similar experiments as those in Fig. 2 on small-scale downstream datasets. The results are shown in Fig. 4. We observe consistent performance improvement as the number of reserved blocks from pre-trained models increases. And the smaller the dataset scale, the more the performance gain from higher layers. It demonstrates that higher layers are still valuable and matter in data-insufficient downstream tasks. Furthermore, we observe comparable performance for the transfer performance of MAE-Tiny and MoCov3-Tiny when only a certain number of lower layers are reserved, while MoCov3-Tiny surpasses when higher layers are further adopted. It indicates the higher layers of MoCov3-Tiny work better than MAE-Tiny on data-insufficient downstream tasks, which is also consistent with our CKA-based analyses shown in Fig. 1, that MoCov3-Tiny learns more semantics at abstract level relevant to recognition in higher layers (high similarity to reference recognition models in higher layers) than MAE-Tiny. And also we conjecture that high-frequency information matters less in these relatively easier tasks.

## 4.2 ATTENTION MAP ANALYSES

The attention maps reveal the behaviors for aggregating information in the attention mechanism, which are computed from the compatibility of queries and keys by dot-product operation. We intro-

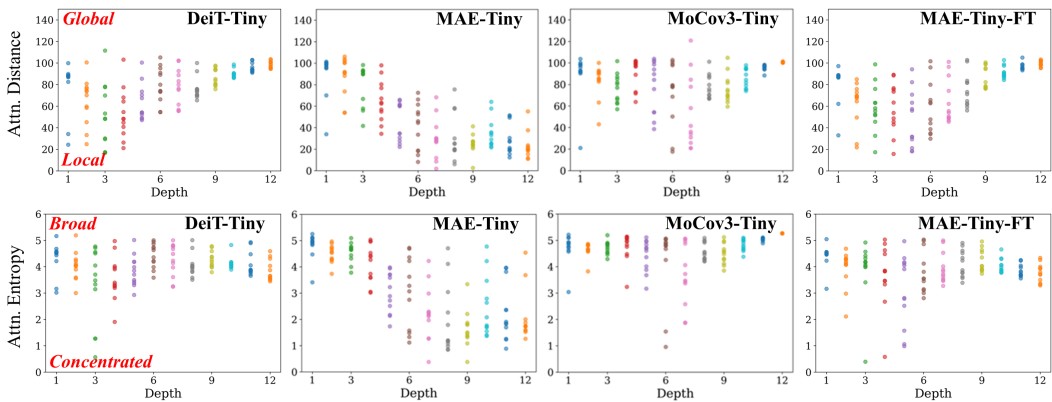

Figure 5: **Attention distance and entropy analyses**. We visualize the averaged attention distance and entropy across all tokens in different attention heads w.r.t. the layer number.

duce two metrics for further analyses on the pre-trained models, *i.e.*, *attention distance* and *attention entropy*. The attention distance for the $j$-th token of $h$-th head is calculated as:

$$\boldsymbol{D}_{h,j} = \sum_i \text{softmax}(\boldsymbol{A}_h)_{i,j} \boldsymbol{G}_{i,j}, \tag{1}$$

where $\boldsymbol{A}_h \in \mathbb{R}^{l \times l}$ is the attention map for the $h$-th attention head, and $\boldsymbol{G}_{i,j}$ is the Euclidean distance between the spatial locations of the $i$-th and $j$-th tokens. $l$ is the number of tokens. And the attention entropy is calculated as:

$$\boldsymbol{E}_{h,j} = -\sum_i \text{softmax}(\boldsymbol{A}_h)_{i,j} \log(\text{softmax}(\boldsymbol{A}_h)_{i,j}), \tag{2}$$

Specifically, the attention distance reveals how much local *vs.* global information is aggregated, and a lower distance indicates that each token focuses more on neighbor tokens. The attention entropy reveals the concentration of the attention distribution, and lower entropy indicates that each token attends to fewer tokens. We analyze the averaged attention distance and entropy across all the tokens in different attention heads, as shown in Fig. 5.

**The pre-training with MAE alters the attention behaviors of the final recognition model little.** First, we compare MAE-Tiny-FT with DeiT-Tiny. The former adopts MAE-Tiny as initialization and then is fine-tuned on IN1K, and the latter is supervised trained from scratch on IN1K. We observe very similar attention behaviors between them. They both have diverse attention heads in lower layers, which aggregate both local and global tokens with both concentrated and broad focus, and more global and broad attention in higher layers. It indicates that the pre-training does not alter the behaviors of the ultimate recognition model much, but provides a better initial state.

**The pre-training with MAE improves the results not by bringing locality inductive bias.** Then, we focus on the attention behaviors of MAE-Tiny. It shows similar patterns to DeiT-Tiny on lower layers, but more distinct patterns on higher layers, which means its higher layers concentrate (low entropy) on local spatial information (low distance). We hypothesize that the behaviors are related to the aim of the pixel reconstruction task in MIM (Masked Image Modeling). Considering only lower layers of MAE-Tiny matter as analyzed previously, we think the pre-training improves the performance not by introducing the locality inductive bias. It indicates that the bias may not be necessary for lightweight models to achieve top performance, which is, however, the key idea of many successful lightweight network architectures (Mehta & Rastegari, 2022; Touvron et al., 2021b; Heo et al., 2021). As for the MoCov3-Tiny, we observe relatively global and broad attention with low diversity in lower layers, which may be not suitable for downstream tasks.

## 5 DISTILLATION IMPROVES PRE-TRAINED MODELS

In this section, we focus on developing a distillation strategy for the top-performing MAE pre-training on lightweight models, to remedy some defects of this strategy. In the previous section,

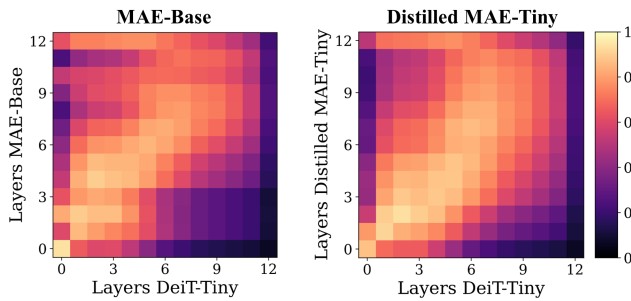
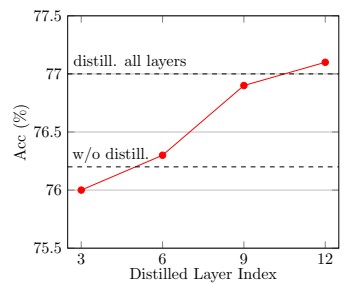

Figure 6: Distillation helps to compress the good representation of the teacher (MAE-Base) to the student, thus the distilled student shows higher similarity to the supervised trained DeiT-Tiny.

Figure 7: Distillation on attention maps of higher layers improves performance most.

we have conjectured that it is hard for MAE to learn good representation relevant to recognition in higher layers, which results in unsatisfactory performance on data-insufficient downstream tasks. A natural question is that can it gain more semantic information by scaling up the models. We further examine a large pre-trained model, MAE-Base (He et al., 2021), and find it achieves a better alignment to DeiT-Tiny, as shown in the left column of Fig. 6. It indicates that *it is possible to extract features relevant to recognition in higher layers for the scaled-up encoder in MAE pre-training.*

These observations motivate us to compress the knowledge of large pre-trained models to tiny ones, *i.e.*, applying knowledge distillation during the pre-training phase for lightweight ViTs. Although it is a common practice to perform distillation to obtain pre-trained compressed language models (Sanh et al., 2019; Jiao et al., 2020; Wang et al., 2020; 2021; Sun et al., 2020; Su et al., 2021), how to apply distillation to obtain better lightweight ViT pre-training under the masked image modeling framework is still unexplored. We fill this gap and propose some useful techniques.

**Distillation methods.** Specifically, a pre-trained MAE-Base (He et al., 2021) is introduced as the teacher network. We adopt the attention-based distillation, which is formulated as follows based on the mean squared error (MSE) between the corresponding layers of the teacher and student:

$$L_{\text{attn}} = \text{MSE}(\boldsymbol{A}^T, \boldsymbol{M}\boldsymbol{A}^S), \tag{3}$$

where $\boldsymbol{A}^T \in \mathbb{R}^{h \times l \times l}$ and $\boldsymbol{A}^S \in \mathbb{R}^{h' \times l \times l}$ refer to the attention maps of the teacher and student with $h$ and $h'$ attention heads, and $l$ is the number of tokens. A learnable mapping matrix $\boldsymbol{M} \in \mathbb{R}^{h \times h'}$ is introduced to align the number of heads. The teacher is also applied only on the same unmasked patches in the encoder as the student during the distillation.

**Distillation on lower or higher layers?** We first examine applying the above layer-wise distillation on which layer contributes to the most performance gain. Though it is a direct way to apply distillation on all corresponding layers of the teacher and student, it actually slows down the training speed. As shown in Fig. 7, only distilling on the attention maps of the last transformer blocks promote the performance most, even surpassing those distilling on all layers or other single lower layers (for the sake of simplicity, we only fine-tune the pre-trained models on IN1K for 100 epochs). It is consistent with the analyses in Sec. 4. Specifically, the lower layers learn good representation themselves during the pre-training with MAE and thus distilling on these layers contributes to marginal improvement, while the higher layers rely on a good teacher to guide them to capture rich semantic features.

**Distillation improves pre-trained models for downstream tasks.** We further evaluate the distilled pre-trained model on several downstream classification tasks (Nilsback & Zisserman, 2008; Parkhi et al., 2012; Maji et al., 2013; Krause et al., 2013; Krizhevsky et al., 2009; Van Horn et al., 2018; Deng et al., 2009) and dense prediction tasks. For simplicity, we only apply distillation on the attention maps of the last layer. The visualization results in Fig. 6 show that the good representation relevant to the recognition of the pre-trained teacher is compressed to the distilled MAE-Tiny. Especially the quality of higher layers is improved. It contributes to better downstream performance on both classification and dense-prediction tasks as shown in Tab. 5, especially on object detection and segmentation tasks, surpassing the supervised pre-training counterpart by a large margin.The above results also support our insight in Sec. 4.1.

Table 5: **Distillation improves downstream performance on classification tasks and object detection and segmentation tasks**. Top-1 accuracy is reported for classification tasks and AP is reported for object detection (det.) and instance segmentation (seg.) tasks.

| Datasets / Init. | Flowers | Pets | Aircraft | Cars | Cifar100 | iNat18 | ImageNet | COCO(det.) | COCO(seg.) |
|---|---|---|---|---|---|---|---|---|---|
| *supervised* DeiT-Tiny | **96.4** | **93.1** | 73.5 | 85.6 | **85.8** | **63.6** | - | 40.7 | 36.5 |
| *self-supervised* MAE-Tiny | 85.8 | 76.5 | 64.6 | 78.8 | 78.9 | 60.6 | 78.0 | 38.9 | 35.1 |
| Distilled MAE-Tiny | 95.2 (+9.4) | 89.1 (+12.6) | **79.2** (+14.6) | **87.5** (+8.7) | 85.0 (+6.1) | **63.6** (+3.0) | **78.4** (+0.4) | **42.7** (+3.8) | **38.2** (+3.1) |

## 6 RELATED WORKS

**Self-supervised learning (SSL)** focuses on different pretext tasks (Gidaris et al., 2018; Zhang et al., 2016; Noroozi & Favaro, 2016; Dosovitskiy et al., 2014) for pre-training without using manually labeled data. Among them, contrastive learning (CL) has been popular and shows promising results on various convolutional networks (ConvNets) (He et al., 2020; Chen et al., 2020; Grill et al., 2020; Caron et al., 2020) and ViTs (Chen et al., 2021a; Caron et al., 2021). Recently, methods based on masked image modeling (MIM) achieve the state-of-the-art on ViTs (He et al., 2021; Bao et al., 2021; Zhou et al., 2022) It has been demonstrated that these methods can scale up well on larger models, while their performance on lightweight ViTs is seldom investigated.

**Vision Transformers (ViTs)** Dosovitskiy et al. (2020) apply a Transformer architecture (a stack of attention modules (Vaswani et al., 2017)) on image patches and show very competitive results in various visual tasks (Touvron et al., 2021a; Liu et al., 2021; Li et al., 2022). The performance of ViTs has been largely improved thanks to better training recipes (Touvron et al., 2021a; Steiner et al., 2021; Touvron et al., 2022). As for lightweight ViTs, most works focus on integrating ViTs and ConvNets (Graham et al., 2021; Heo et al., 2021; Mehta & Rastegari, 2022; Chen et al., 2021b; Yan et al., 2021), while few focus on how to optimize the networks.

**Knowledge Distillation** is a mainstream approach for model compression(Buciluǎ et al., 2006), in which a large teacher network is trained first and then a more compact student network is optimized to approximate the teacher (Hinton et al., 2015; Romero et al., 2014; Shen & Xing, 2022). Touvron et al. (2021a) achieves better accuracy on ViTs by adopting a ConvNet as the teacher. With regard to the compression of the pre-trained networks, some works (Sanh et al., 2019; Jiao et al., 2020; Wang et al., 2020; 2021; Sun et al., 2020; Su et al., 2021) attend to distill large-scale pre-trained language models. In the context of computer vision, a series of works (Fang et al., 2020; Abbasi Koohpayegani et al., 2020; Choi et al., 2021; Shen et al., 2021) focus on transferring knowledge of large pre-trained networks based on CL to lightweight ConvNets. There are few works focusing on improving the quality of lightweight pre-trained ViTs based on MIM by distillation thus far.

## 7 DISCUSSIONS

**Limitations** Our study is restricted to classification tasks and some dense-prediction tasks, *e.g.*, object detection and segmentation. We leave the exploration of more tasks for further work.

**Conclusions** We investigate the self-supervised pre-training of lightweight ViTs, and demonstrate the usefulness of the advanced lightweight ViT pre-training strategy in improving the performance of downstream tasks. Some properties about the pre-training are revealed, *e.g.*, these methods fail to benefit from large-scale pre-training data, and show more dependency on the downstream dataset scale. We also present some insights on what matters for the downstream performance with pre-training by analyzing the layer representation and attention map. They may indicate potential future directions in improving pre-training on lightweight models, the value of which has also been demonstrated as it guides the design of our proposed distillation strategy and helps to achieve much better downstream performance. We expect our research may provide useful experience and advance the study of self-supervised learning on lightweight ViTs.

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
