# OpenReview forum: "A Closer Look at Self-supervised Lightweight Vision Transformers"
_ICLR.cc/2023/Conference — Submitted to ICLR 2023_

### Official Review · Reviewer_nNkX · 2022-10-21

**Confidence:** 4
**Correctness:** 4
**Technical Novelty And Significance:** 2
**Empirical Novelty And Significance:** 2
**Recommendation:** 6

**Clarity, Quality, Novelty And Reproducibility:**

Clarity
The paper is fairly clear, although there are times I wish ablations had been kept separate. For instance, "we adopt the layer-wise attention-based distillation" and then two paragraphs later "For simplicity, we only apply distillation on the attention maps of the last layer". This causes some confusion and it's easier for the reader if you present one main method, and leave ablations to separate sections.

Quality
The analysis and results are sound and of good quality. The experimental evaluation is varied and relevant.

Novelty
As for novelties, this paper mostly presents analysis and the final distilled MAE uses standard methods. The novelty is in applying these to more lightweight ViTs.

Reproducibility
As for reproducibility, it is unclear if source code is forthcoming.

Suggestions:
- Adding a training data column to tables 4 and 6  would help to make it clear that DeiT used labels and the other methods did not. Having to go through the text to realize this takes time.
- Why not include DeiT in table 6 as well? The distilled MAE actually outperforms the supervised method on several benchmarks. This is significant and should be highlighted.

**Strength And Weaknesses:**

Strengths:
- This is a pretty in depth analysis of the difference between a MAE- and a MoCo-based self-supervised ViT.
- The distillation of the MAE-Tiny from MAE-Base leads to significant improvements over the MAE-Tiny without teacher guidance. The final results compared to supervised pretraining is strong and a significant achievement.

Weaknesses:
- The paper mostly presents analysis and new applications of old techniques. It is pretty light on novelties though.
- Not all analysis is tied back to experimental results, which means it's not clear if we can draw actionable conclusions from it. If the analyses had led to more actionable changes that resulted in improvements, the case would have been stronger.

**Summary Of The Paper:**

This paper presents an analysis of two self-supervised training methods for lightweight vision transformers. The methods are MAE (a masking method) and MoCoV3 (a contrastive method). It shows that although MAE appears stronger when training (w/o labels) and evaluating on ImageNet, the roles reverse on downstream tasks. It analyzes the differences of these two networks and compares them a supervised method (DeitT), by looking at layer-by-layer similarity metrics, as well as analysis of the locality vs. globality of the attention maps. Finally it proposes to use distillation from a large-scale MAE model to improve the tiny MAE. It does this with significant improvements to downstream tasks.

**Summary Of The Review:**

The papers presents a lot of analysis around lightweight ViTs, and shows the efficacy of student/teacher distillation for mask-based self-supervised pre-training. The paper presents few novelties in terms of technique and method, but the quality and novelty of the analysis and strength of final distilled results are solid contributions. The topic is of broad interest, so many could benefit from this work. Balancing the lack of novelties with the potential benefits it is close to borderline for me. However, I do lean in favor of acceptance.

---

> ### Author Response · Authors · 2022-11-18
> **Author Response to Reviewer nNkX**
>
> Thanks for your review and feedback. We include detailed responses below.
>
> **About novelty**: As an analysis paper, our work mainly focuses on:
> 1. exploring the great potential of the pre-training on lightweight models, one of the most hopeful ways towards top-performing models by optimizing the training schemes, based on which our achieved (nearly) top results with vanilla ViTs (Tab 3) may draw the research attention from designing new architectures (the mainstream of current ViT study) to the less studied optimization strategies with pre-training;
> 2. most importantly, giving some new insights that hide behind the achieved results to help inform future directions in pre-training on lightweight models, and the value of it has also been demonstrated by the success of our proposed distillation strategy, which follows the guidance of our analyses and achieves better results, especially on small downstream classification datasets and object detection & segmentation tasks.
>
> **About actionable conclusions**: Our analyses point out several intriguing defects of applying existing SSL methods to lightweight ViTs, which we deem to be valuable themselves. Proposing solutions to all of these problems is out of the scope of the paper. Nonetheless, we think our analyses have the following inspiration for future works on self-supervised pre-trained lightweight models:
> - Improving the quality of higher layers of MAE may boost downstream performance on data-insufficient tasks. Our distillation-based method is exactly the one that follows this instruction and achieves remarkable improvement.
> - Avoiding over-spatially-smoothed feature map and preserving more useful high-frequency signals may boost downstream performance for CL-based methods, especially when data is sufficient.
>
> **About writing and presentation of Table 4 and 6**: We have polished our writing carefully, and avoided ambiguous expressions. For instance, we have revised the presentation of our distillation-based method in Sec. 5 and also revised Table 4 and Table 6 (Table 5 in the revision) following your suggestions. Please refer to the revised version for more details.
>
> **About reproducibility**: Our code is attached in the revised supplementary material.

---

> > ### Comment · Reviewer_nNkX · 2022-12-12
> > **Feedback acknowledgement**
> >
> > Thank you for the clarifications. After reading the feedback and the other reviews (which I think are fair assessments and I do not disagree with, even though my score was a bit more favorable), I will neither lower my rating, nor stick my neck out to champion it. Best of luck continuing this line of work!

---

> > > ### Author Response · Authors · 2022-12-13
> > > **Many thanks for the comment**
> > >
> > > As suggested, we will continue this line of work and do have the plan of demonstrating its value in the real robot vision and autonomous systems.

---

### Official Review · Reviewer_UM5C · 2022-10-23

**Confidence:** 4
**Correctness:** 3
**Technical Novelty And Significance:** 2
**Empirical Novelty And Significance:** 3
**Recommendation:** 5

**Clarity, Quality, Novelty And Reproducibility:**

This paper is well written and easy to understand. This reveals some secrets behind SSL on visual recognition using lightweight ViT models. The novelty is ok but not significant enough for publication.

**Strength And Weaknesses:**

Strength
- Analyzing the performance of lightweight ViTs in self-supervised learning manners is important in that in most cases it is not appropriate to use large-scale models but only tiny-sized ones. This paper clearly shows how lightweight ViTs behave under different settings and evaluates the pretrained models on downstream tasks.
- Authors show that low-level layers are more important than high-level layers after pretraining. This is an important signal for the development of SSL methods in future.
- Distilling the knowledge from a large model to lightweight ones is an interesting topic. This paper provides an effective way to do so and analyze how to do distillation helps more for the lifting the model performance.

Weaknesses
- Only two SSL methods are selected for presentation. One is MOCOv3 and the other one is MAE. It would be better if two methods are selected for explanation for each type of SSL method. The conclusion would be more convincing.
- From Table 3, we see that the pre-training benefits little from large-scale data.  With only 10% of the ImageNet 1k data are provided, the classification performance after finetuning is already good. I suppose there are some explanations on the reasons but obviously there is not.
- The analysis is interesting. I am looking forward to taking some message about how to design better network architectures or how to develop more advanced self-supervised learning methods.

**Summary Of The Paper:**

This paper aims to analyze how the lightweight ViTs (e.g., ViT-tiny and DeiT-tiny) perform when using self-supervised pretraining methods. By conducting a variety of experiments, the authors show that: 1) MIM-based methods, like MAE, helps more than contrastive learning based methods, like MOCOv3; 2) Low-level layers matter more than high-level layers when sufficient data for finetuning is available; 3) KD methods help improve the representative ability of high-level layers.

Previous works mostly focus on large-scale models, which often contain more than 80M parameters. In contrast, these paper aims to reveal how lightweight ViTs behave on both ImageNet and some downstream tasks, which makes this paper much more different than previous works on self-supervised methods.

**Summary Of The Review:**

The intention of this paper is interesting. There are no relevant papers reporting similar conclusions. Though the novelty of this paper is not significant enough, regarding the thorough experiments that have been done, I give a score of 5 at this moment. I would like to lift the rating if the authors can more clearly explain the significance of the contributions.

---

> ### Author Response · Authors · 2022-11-18
> **Author Response to Reviewer UM5C**
>
> Thanks for your review and feedback. We include detailed responses below.
>
> **About more SSL methods for presentation**: MAE is one of the most simple SSL methods that achieve SOTA performance based on MIM, and MoCov3 also achieves SOTA performance among CL methods with rather simple architecture. As you suggested, we have recently further examined some other SSL methods on lightweight ViTs to validate the generalization of our analyses, with MIM-based SimMIM and CL-based DINO involved. We find that the methods based on MIM show similar behaviors and so do those based on CL. We have added these new analyses in Appendix B.5 of the revised supplementary material ( Please also see https://ibb.co/F7WVW7f for a quick check).
>
> **About explanations on performance based on various pre-training data scale**: We think the tiny model scale restricts the model capacity, which limits the ability of benefiting from large-scale pre-training data. We leave further exploration for future work and expect our observations could encourage more researchers to explore this topic.
>
> **About some "take-away" messages on**
>
> - **designing better network architectures**: Our experimental results reveal that: designing new architectures (the mainstream of current lightweight ViT study) is not the only way toward top-performing lightweight models, whereas, optimizing the training schemes is also of vital importance. We argue that the neglect of the training strategy may lead to a misjudgment of the power of the existing model architectures. One example is that we achieve (nearly) top performance on ImageNet with vanilla ViT architecture, which is previously thought to be not suitable for lightweight regimes, and surpass most ViT derivatives with delicate design by only adopting proper pre-training.
>
> - **developing more advanced SSL methods**: Our experimental results reveal that: current pre-training approaches can not achieve consistent superiority on various downstream tasks, and our analyses also put forward some proposals for SSL improvement, e.g., improving the quality of higher layers of MAE may boost downstream performance on data-insufficient tasks. Our distillation-based pre-training method can also be viewed as one of the examples that follows the instruction of our analyses and achieves remarkable improvement.
>
> We hope the above responses can address your concern about the significance of our contributions and respectfully ask you to consider raising the score in support of accepting this paper.

---

> > ### Comment · Reviewer_UM5C · 2022-11-25
> > **Thanks for the feedback**
> >
> > After reading the reviews and feedback, I hold the same opinion with Reviewer PEwP that the novelty of this paper is not significant enough for publication in ICLR. I learn little based on the findings presented in the paper. However, regarding the thoughrout experiments, I keep my original rating unchanged.

---

> > > ### Author Response · Authors · 2022-12-13
> > > **Thanks for the comment**
> > >
> > > As an analysis paper, our work is dedicated to providing the community with the in-depth analyses throughout the experiments, and we have made many efforts to conduct them.

---

### Official Review · Reviewer_PEwP · 2022-10-26

**Confidence:** 5
**Correctness:** 2
**Technical Novelty And Significance:** 1
**Empirical Novelty And Significance:** 1
**Recommendation:** 3

**Clarity, Quality, Novelty And Reproducibility:**

The clarity of this paper is qualified. Novelty and originality are somewhat limited. Since this paper did not introduce any new and concrete approach, reproducibility is not applicable.

**Strength And Weaknesses:**

### Strengths:

   - It is interesting to explore suitable methods for lightweight vision transformers or other efficient models in the self-supervised learning manner.

   - This paper provided extensive experiments on ImageNet pre-training and multiple downstream datasets and tasks, such as classification, object detection, and segmentation.


### Weaknesses:

   - Though this is an empirical paper, the novelty and originality in it are fairly limited, as well as the significance and contribution which are also not strong. The observations that tiny models fail to benefit from large-scale pre-training data, and rely more on the downstream dataset scale are a little bit straightforward and not surprising. The use of knowledge distillation for self-supervised learning on lightweight models also has been proposed for a long time, e.g., on low-bit efficient models [1] and mobile-level models [2].

[1] Shen, Z., Liu, Z., Qin, J., Huang, L., Cheng, K. T., & Savvides, M. (2021). S2-bnn: Bridging the gap between self-supervised real and 1-bit neural networks via guided distribution calibration. In Proceedings of the IEEE/CVF Conference on Computer Vision and Pattern Recognition (pp. 2165-2174).

[2] Fang, Zhiyuan, Jianfeng Wang, Lijuan Wang, Lei Zhang, Yezhou Yang, and Zicheng Liu. "SEED: Self-supervised Distillation For Visual Representation." In International Conference on Learning Representations. 2021.

   - Some statements in this paper are not well supported, such as “lower layers of the pre-trained models matter more than higher ones if sufficient downstream data is provided, while higher layers matter in data-insufficient downstream tasks.” I think a better and fair comparison design is crucial and also necessary for this argument. The current experiments for this part are not rigorous to prove it.

   - The writing and organization of this paper can also be improved. For instance, it’s not clear to me why Table 1 is located in the early part of the paper. I did not get much information from it and do not know what the insight of this table is.

   - Overall, this paper seems a little bit incremental without providing new conclusions or discoveries over previous literature.



**Summary Of The Paper:**

This is an empirical paper focusing on exploring the task of self-supervised learning on lightweight vision transformers (ViTs). In this paper, the authors observed several discoveries, such as: the tiny model fails to benefit from large-scale pre-training data and shows inferior performance on data-insufficient downstream tasks. They further proposed a distillation strategy during pretraining to improve the representation ability of compact ViTs. Experiments are conducted on ImageNet pre-training and multiple downstream tasks and datasets.

**Summary Of The Review:**

Overall, this is an empirical paper with some trivial observations which are not well supported by the experiments and some are even not new. Also, the organization and writing can be improved significantly in this paper. Thus, I tend to reject it.

---

> ### Author Response · Authors · 2022-11-18
> **Author Response to Reviewer PEwP**
>
> Thanks for your review and feedback. We include detailed responses below.
>
> **About our novelty, originality and contribution**: As an analysis paper, our work mainly focuses on:
> 1. exploring the great potential of the pre-training on lightweight models, one of the most hopeful ways towards top-performing models by optimizing the training schemes, based on which our achieved (nearly) top results with vanilla ViTs (Tab 3) may draw the research attention from designing new architectures (the mainstream of current ViT study) to the less studied optimization strategies with pre-training;
> 2. giving some new insights that hide behind the achieved results to help inform future directions in pre-training on lightweight models, the value of which has also been demonstrated as it guides the design of our proposed distillation strategy and helps to achieve much better downstream performance, especially on data-insufficient classification tasks, and object detection and segmentation tasks.
>
> We believe our work has significant and solid contributions to the literature with a number of new conclusions or discoveries over previous literature. For instance, Reviewer `UM5C` recognizes this paper *much more different than previous works* and acknowledges that *there are no relevant papers reporting similar conclusions*. Reviewer `nNkX` also thinks that *the quality and novelty of the analysis and strength of final distilled results are solid contributions*, and *many could benefit from this work*.
>
> **About knowledge distillation**: Applying knowledge distillation is one of the most popular and promising strategies to achieve good lightweight pre-trained models, whereas, different approaches are demanded for different SSL methods and networks. The mentioned works [1] and [2] focus on improving binary pre-trained networks and lightweight CNN respectively based on contrastive learning with knowledge distillation, while our work focuses on improving lightweight pre-trained ViTs based on MIM, which is seldom investigated and obviously different from the previous works. We've also clarified the relation of our work to them in Sec. 6. Thanks for pointing out some missing related works. They have been appended to our references along with some other recent works on knowledge distillation in the revision.
>
> **About a better and fair comparison design** for our statements: In Sec 4.1, we present not only analyses based on layer similarity, but also solid experimental analyses as shown in Fig. 2, which shows that by only adopting several lower layers of MAE-Tiny as initialization, significant performance gains are achieved on ImageNet. As you suggested, we have recently conducted similar experiment with MoCov3 and further extended to other small-scale downstream datasets. As shown in https://ibb.co/4fWrFsn, the results show that:
>
> - On ImageNet, adopting only lower layers of MAE and MoCov3 as initialization achieves significant performance improvement, while further utilizing higher layers leads to marginal gain (for MAE) or even degradation (for MoCov3), which demonstrates that lower layers matter more than higher ones for both MIM-based MAE and CL-based MoCov3 if sufficient downstream data is provided.
> - On data-insufficient downstream tasks (e.g., Cars, Pets and CIFAR100), we observe consistent improvement as the number of reserved blocks from pre-trained models increases. And the smaller the dataset scale, the more the performance gain from higher layers. It demonstrates that higher layers matter in data-insufficient downstream tasks.
>
> The above analyses have been added in Sec. 4.1 of our revised version.
>
> **About the writing and organization**: We have polished our writing carefully. As for Table 1, we think that constructing experimental benchmarks is the first step to investigate this new topic, which is exactly the main content of Sec. 3. Table 1 contains transfer performance on ImageNet for lightweight ViTs pre-trained with various approaches, and reveals that proper pre-training by us also achieves remarkable downstream performance improvement, which establishes the practical significance of our study and is also the prerequisite of our analyses in Sec. 4. Thus, we still insist that Table 1 is properly located.

---

### Decision · Program_Chairs · 2023-01-20

**Decision:**

Reject

**Justification For Why Not Higher Score:**

This paper explores the task of self-supervised learning on lightweight vision transformers (ViTs) that fits well with our venue.  However, after discussion and despite a response from the authors that addressed some of the concerns raised during the reviews, all reviewers agreed that the level of contribution was too limited in terms of novelty.  Reviewer nNkX's score was a bit more favorable but decided not to champion the paper.

The AC hence suggests that the paper is not yet ready for publication with its current version and encourages the authors to revise the paper in light of the reviewers' comments and submit it to a future venue.

**Justification For Why Not Lower Score:**

N/A

**Metareview: Summary, Strengths And Weaknesses:**

This paper explores the task of self-supervised learning on lightweight vision transformers (ViTs).  What makes the paper different from existing works (often contain more than 80M parameters. ) is that it aims to reveal how lightweight ViTs behave on both ImageNet and some downstream tasks. The summary of strengths and weaknesses as I summarize from the reviewers' feedback and the paper are:


Strengths
---------------
 - Reviewer PEwP and Reviewer UM5C:  "It is interesting to explore suitable methods for lightweight vision transformers or other efficient models in the self-supervised learning manner."

- Reviewer UM5C: "Authors show that low-level layers are more important than high-level layers after pretraining. This is an important signal for the development of SSL methods in future."

-  Reviewer UM5C and Reviewer nNkX:  "This paper provides an effective way to do so and analyze how to do distillation helps more for the lifting the model performance."

- Reviewer PEwP and Reviewer nNkX:  "This paper provided extensive experiments on ImageNet pre-training and multiple downstream datasets and tasks, such as classification, -object detection, and segmentation." "The distillation of the MAE-Tiny from MAE-Base leads to significant improvements over the MAE-Tiny without teacher guidance. "


Weaknesses
---------------
- Reviewer PEwP:  "Though this is an empirical paper, the novelty and originality in it are fairly limited, as well as the significance and contribution which are also not strong. "
- Reviewer UM5C: "Only two SSL methods are selected for presentation. One is MOCOv3 and the other one is MAE. It would be better if two methods were selected for explanation for each type of SSL method. The conclusion would be more convincing."
- Reviewer UM5C: From Table 3, we see that the pre-training benefits little from large-scale data.
- Reviewer nNkX:  "It is pretty light on novelties. "
- Reviewer nNkX:  Not all analysis is tied back to experimental results, which means it's not clear if we can draw actionable conclusions from it. If the analyses had led to more actionable changes that resulted in improvements, the case would have been stronger"